# Inflation and Risky Investments

**Hannu Laurila * and Jukka Ilomäki**

Faculty of Management and Business, Tampere University, FI-33014 Tampere, Finland; jukka.ilomaki@tuni.fi
* Correspondence: hannu.laurila@tuni.fi

**Abstract:** The paper uses a Walrasian two-period financial market model with informed and uninformed constant absolute risk averse (CARA) rational investors and noise traders. The investors allocate their initial wealth between risky assets and risk-free fiat money. The analysis concentrates on the effects of decreasing value of money, or inflation, on the rational investors' behavior and the asset market. The main findings are the following: Inflation does not affect the informed investors' prediction coefficient but makes that of the uninformed investors diminish. Inflation does not affect rational investors' risk but makes the asset price more sensitive to fundament-based and sentiment-based shocks. Inflation changes the market price of the risky asset rise; while it has no effects on the informed investors' demand of the risky asset, it does affect the uninformed investors' demand. Finally, inflation makes the asset market more volatile.

**Keywords:** Constant Absolute Risk Averse (CARA) utility function; information; negative interest rates; volatility

---

## 1. Introduction

Zero or negative real interest rates have recently been common in many countries, including the USA, Japan, Switzerland, Norway, Sweden and the Eurozone. Caballero and Farhi (2018) discuss the issue and call it the "safety trap". It causes a shortage in safe assets and reduces the wealth of risk averse investors, who tend to allocate a part of their investments in riskless assets. If the investors' wealth is mitigated, then also their consumption falls so that aggregate consumption in the economy is reduced (see also Caballero 2008; Bernanke et al. 2011).

Viewing from the supply side, Summers (2013) notes that the safety trap may dampen output expansion even if there is a boom in the stock market. The seminal concept of liquidity trap (Keynes 1936; Dominguez et al. 1998; Eggertsson and Woodford 2003) complements the picture by telling that monetary policy becomes ineffective when nominal interest rates get very low.

Caballero and Farhi (2018) also argue that a fall in the risk-free rate to zero and below makes investors' risk premiums rise because of the malign effects on output. The authors warn that the effect can be persistent and similar to the secular stagnation hypothesis (see Hansen 1939; Summers 2014). They suggest that the investors' risk premium should be lowered by policy and claim that the early quantitative easing in the US subprime crisis and the outright monetary transactions in the Eurozone in late 2012 were apt examples of fight against the safety trap. Yet, as these unconventional monetary policy measures mean money inflation, a common perception is that it leads to price inflation.

Thus, while the phenomenon of zero or negative real interest rates is mainly a monetary policy issue, it is clearly relevant to private savers and investors, who make long-term decisions about wealth accumulation and consider risk in their investments. In common sense, this means decisions of holding either risk-free or risky assets. The basic setting is the decision between liquid or non-liquid wealth, that is, between fiat money in paper form or in current accounts and financial assets.

Nowadays, holding of paper money suffers from modest but still noteworthy price inflation, and current accounts pay (close to) zero nominal interest rates while the commercial banks charge substantial account maintenance fees. Thus, the real value (or purchasing power) of paper money depends on inflation, and the value of current account money depends on factual nominal rates and inflation. The fact that the factual nominal interest rates for current account money are very low has inspired Eggertsson et al. (2019) to argue that the existence of paper money prevents them from falling below zero.

In this paper, we take fiat money as the risk-free alternative to risky investments and examine the effects of the depreciation of its gross yield. That is, our key questions concern the effects of price inflation on rational investors' behavior and further on the asset market. We apply a Walrasian two-period framework consisting of asymmetrically informed constant absolute risk averse (CARA) rational investors, and sentiment-based noise traders.

The main findings are that while inflation does not affect the risk of rational investors, it makes them more sensitive to market information thus affecting the market price of the risky assets. The better-informed investors' demand of risky assets remains unaffected, but the less informed investors are induced to change their demand. The market price of the risky asset is affected, and volatility in the financial market is increased.

As far as we know, our approach has not been previously used to investigate these issues, and we therefore pay close attention to the formalization of the model and to the derivation of the results. For the same reason, there are only few references on which we could directly reflect our findings.

The paper proceeds as follows. Section 2 presents the basic model and the financial market equilibrium. Section 3 presents the analysis of the effects of a marginal rise in inflation on the investment behavior of CARA rational investors with due consequences to the financial market. Section 4 concludes and discusses the findings.

## 2. The Model

Our Walrasian financial market model (see Grossman and Stiglitz 1980; Admati 1985; Mendel and Shleifer 2012; Ilomäki and Laurila 2018a; Chang et al. 2019) includes rational constant absolute risk averse (CARA) investors, who allocate their investments between risky and riskless assets. The rational investors are divided into "informed" and "uninformed" investors so that $0 < \mu < 1$ of them are informed and the rest $1$-$\mu$ are uninformed. In addition, there are correlated noise traders who trade upon pure sentiment. In the two-period model, trading takes place in the first period, and consumption of the accumulated wealth happens in the second period.

The risky asset is stocks that pay a dividend per share like: $D \sim N(\overline{D}, \sigma_D^2)$, where $\widetilde{D}$ is the actual dividend, $\overline{D}$ is a parameter that represents the unconditional expectation of the random variable $\widetilde{D}$ and $\sigma_D^2$ is the variance of $\widetilde{D}$. The informed investors observe a noisy signal $\widetilde{s}$ about the fundamental value of the risky asset so that: $\widetilde{s} = \left[\widetilde{D} - \overline{D}\right] + \varepsilon$ with $\varepsilon \sim N(0, \sigma_\varepsilon^2)$, where $\sigma_\varepsilon^2$ is the variance of the noise error $\varepsilon$. The uninformed investors infer $\widetilde{D}$ from the current market price $P$ of the asset, and the noise traders utilize their correlated sentiment on $\widetilde{D}$.

The riskless investment alternative is fiat money, either in paper form or deposited in current accounts. The argument by Eggertsson et al. (2019) that the zero nominal interest rate on paper money presents the lower bound for the factual rates for electronic money implies that, at the optimum, the net nominal rate of return on both must be zero. Thus, denoting the gross yield from risk-free fiat money by $r$ and assuming $0 < r \leq 1$ means that the net nominal rate of return is zero at $r = 1$ and turns negative because of inflation.

The market allocation of the risky asset reads:

$$\mu x_I + (1 - \mu)x_U + \widetilde{N} = A. \tag{1}$$

In (1), $x_I$ and $x_U$ denote the net demand of informed and uninformed investors, respectively. $\widetilde{N}$ is the noise traders' net demand of the risky asset and $\widetilde{N} \sim N(0, \sigma_N^2)$, where $\sigma_N^2$ is the variance of the random variable $\widetilde{N}$. $A$ is the available fixed amount of the risky asset.

The informed investors form their expectation of $\widetilde{D}$ based on their private signal $\widetilde{s}$:

$$E\left[\widetilde{D}\big|\widetilde{s}\right] = \overline{D} + \beta[\widetilde{s}]. \tag{2}$$

In (2), the informed investors' statistical prediction coefficient $\beta$ presents the probability to get accurate information from the signal in forming the expectation on $\overline{D}$. Obeying the OLS method and recalling that: $\widetilde{s} = \left[\widetilde{D} - \overline{D}\right] + \varepsilon$ with $\widetilde{D} \sim N(\overline{D}, \sigma_D^2)$ and $\varepsilon \sim N(0, \sigma_\varepsilon^2)$, the prediction coefficient comes from: $\beta = \frac{\text{cov}\left[(\widetilde{D}-\overline{D})+\varepsilon, \widetilde{D}\right]}{\text{var}\left[(\widetilde{D}-\overline{D})+\varepsilon\right]}$ and reads:

$$\beta = \frac{\sigma_D^2}{\sigma_D^2 + \sigma_\varepsilon^2}. \tag{3}$$

Note that (3) gives $\beta$ in terms of underlying parameters and $0 < \beta < 1$. The variance of the informed investors' prediction error then comes from: $\sigma_I^2 = \frac{\text{var}(\widetilde{D}-\overline{D})\text{var}(\varepsilon)}{\text{var}\left[(\widetilde{D}-\overline{D})+\varepsilon\right]}$ and reads:

$$\sigma_I^2 = \frac{\sigma_D^2 \sigma_\varepsilon^2}{\sigma_s^2} = \frac{\sigma_D^2 \sigma_\varepsilon^2}{\sigma_D^2 + \sigma_\varepsilon^2}. \tag{4}$$

The uninformed investors' expectation on $\widetilde{D}$ arises from:

$$E\left[\widetilde{D}\big|P\right] = \overline{D} + \gamma\left[P - \overline{P}\right], \tag{5}$$

where $\gamma$ is their prediction coefficient. In order to compute the variance of the uninformed investors' prediction error, the formation of the risky asset's market price must be known.

A tentative assumption is that the market price $P$ depends on the private signal $\widetilde{s}$ and the net demand of the noise traders $\widetilde{N}$ like: $P = z + b\widetilde{s} + c\widetilde{N}$, where z denotes the constant term and parameters $b$ and $c$ refer to rational investors' sensitivity to exogenous shocks in $\widetilde{s}$ and $\widetilde{N}$, respectively. Recalling that $\widetilde{s} = \left[\widetilde{D} - \overline{D}\right] + \varepsilon$ and $\widetilde{N} \sim N(0, \sigma_N^2)$, the expected value of both $\widetilde{s}$ and $\widetilde{N}$ is zero.

Since $\overline{P}$ in (5) is the unconditional expectation of $P$ that is the average price, it represents the constant so that: $z = \overline{P}$. Thus, the tentative assumption on market price formation reads:

$$P = \overline{P} + b\widetilde{s} + c\widetilde{N}, \tag{6}$$

where the constant $\overline{P}$ and the sensitivity parameters $b$ and $c$ are unknown until the model is closed. Using formula (6) for $P$ in (5), the uninformed investors' expectation of $\widetilde{D}$ reads:

$$E\left[\widetilde{D}\big|P\right] = \overline{D} + \gamma\left[b\widetilde{s} + c\widetilde{N}\right]. \tag{7}$$

In (7), the uninformed investors' prediction coefficient $\gamma$ is calculated like: $\gamma = \frac{b\,\text{cov}(\widetilde{s}, \widetilde{D})}{\text{var}(b\widetilde{s}) + \text{var}(c\widetilde{N})}$ which can be written as:

$$\gamma = \frac{b\sigma_D^2}{b^2(\sigma_D^2 + \sigma_\varepsilon^2) + c^2\sigma_N^2}. \tag{8}$$

The variance of the uninformed investors' prediction error then comes from: $\sigma_U^2 = \frac{\text{var}(\widetilde{D})\text{var}(b\varepsilon) + \text{var}(\widetilde{D})\text{var}(c\widetilde{N})}{var(b\widetilde{s}) + \text{var}(c\widetilde{N})}$ which turns to:

$$\sigma_U^2 = \theta\sigma_D^2, \tag{9}$$

where:

$$\theta = \frac{b^2\sigma_\varepsilon^2 + c^2\sigma_N^2}{b^2(\sigma_D^2 + \sigma_\varepsilon^2) + c^2\sigma_N^2}. \tag{10}$$

Note that $0 < \theta < 1$ regardless of the values of the so far unknown parameters $b$ and $c$.

Next, construct the financial market equilibrium described by formula (1), which necessitates derivation of the informed and uninformed investors' demand functions.

Taken that the expected logarithmic returns of the risky asset are normally distributed, the expected consumption utility of an informed CARA investor can be written as follows:

$$E\left[u(\widetilde{c_I}|\widetilde{s})\right] = -e^{-\{x_I E[\widetilde{D}|\widetilde{s}] + [a_0 - x_I P]r - \frac{x_I^2\sigma_I^2}{2}\}}, \tag{11}$$

where $x_I$ denotes the informed investor's demand of the risky asset, and $a_0$ is the initial endowment of money. Taking the first-order maximum condition against $x_I$ produces: $E\left[\widetilde{D}|\widetilde{s}\right] - rP - x_I\sigma_I^2 = 0$, and solving for $x_I$ gives:

$$x_I = \frac{E\left[\widetilde{D}|\widetilde{s}\right] - rP}{\sigma_I^2}. \tag{12}$$

Equation (12) says that the informed investor's demand for the risky asset equals the expected gain from buying the risky asset divided by the variance of the prediction error $\sigma_I^2$, which is spelled out in terms of underlying parameters in (4). The variance of the prediction error can be regarded as the risk of a constant absolute risk averse investor, whose CARA coefficient equals 1 by assumption.

Likewise, an uninformed investor's expected utility function reads:

$$E[u(\widetilde{c_U}|P)] = -e^{-\{x_U E[\widetilde{D}|P] + [a_0 - x_U P]r - \frac{x_U^2\sigma_U^2}{2}\}}, \tag{13}$$

and solving from the first order maximum condition produces:

$$x_U = \frac{E\left[\widetilde{D}|P\right] - rP}{\sigma_U^2} \tag{14}$$

as the uninformed investor's demand for the risky asset. Again, $\sigma_U^2$ can be interpreted as the uninformed CARA investor's risk.

The market allocation (1) can now be constructed by plugging the formulas (2) and (7) into (12) and (14), respectively. This produces:

$$\mu\left[\frac{\overline{D} + \beta\widetilde{s} - r(\overline{P} + b\widetilde{s} + c\widetilde{N})}{\sigma_I^2}\right] + (1 - \mu)\left[\frac{\overline{D} + \gamma(b\widetilde{s} + c\widetilde{N}) - r(\overline{P} + b\widetilde{s} + c\widetilde{N})}{\sigma_U^2}\right] + \widetilde{N} = A. \tag{15}$$

In the Walrasian equilibrium, the sum of the constants on the left-hand side of (15) must equal the fixed amount of assets $A$ on the right-hand side, and the sum of the respective coefficients connected to $\widetilde{s}$ and $\widetilde{N}$ must be zero. Hence, the so far unknown $\overline{P}$, $b$ and $c$ can now be solved as:

$$\overline{P} = \frac{1}{r}\left[\overline{D} - \frac{\sigma_I^2\sigma_U^2}{\mu\sigma_U^2 + (1 - \mu)\sigma_I^2}A\right], \tag{16}$$

$$b = \frac{\mu\beta\sigma_U^2}{r\mu\sigma_U^2 + (1 - \mu)(r - \gamma)\sigma_I^2}, \tag{17}$$

$$c = \frac{\sigma_I^2\sigma_U^2}{r\mu\sigma_U^2 + (1 - \mu)(r - \gamma)\sigma_I^2}. \tag{18}$$

Equation (16) describes the constant and says that the unconditional expectation of $P$ that is the average price $\overline{P}$ equals the discounted value of the net sum of the expected dividend, and available assets $A$ multiplied by the market risk term. Comparing to the conventional pricing formula $\overline{P} = \frac{\overline{D}}{r}$, Equation (16) adds the risk term implying that very big values of $A$ may make the sign of the constant negative. This means that the average market price may become negative, which is due to the artifact of the CARA utility function. However, a plausible assumption is that $\overline{P} > 0$.

Equations (17) and (18) present the sensitivity parameters $b$ and $c$ and show they depend on $\gamma$ and $\theta$, which depend again on $b$ and $c$ by (8) and (10), respectively. The interdependency must be examined.

Start by assessing $\theta$ as it appears by (9) through $\sigma_{U}^2$ in all the formulas (16)–(18). Divide $b$ from (17) by $c$ from (18), use (3) and (4), and get: $\frac{b}{c} = \frac{\mu\beta}{\sigma_I^2} = \frac{\mu}{\sigma_\varepsilon^2}$. Taking squares and manipulating produces: $b^2 = \frac{c^2\mu^2}{\sigma_\varepsilon^4}$, and plugging this into (10) gives: $\theta = \frac{(\mu^2\sigma_\varepsilon^2/\sigma_\varepsilon^4)+\sigma_N^2}{(\mu^2\sigma_s^2/\sigma_\varepsilon^4)+\sigma_N^2}$. Multiplying by $\frac{\sigma_\varepsilon^4}{\sigma_\varepsilon^4}$ and manipulating yields:

$$\theta = \frac{\mu^2\sigma_\varepsilon^2 + \sigma_\varepsilon^4\sigma_N^2}{\mu^2(\sigma_D^2 + \sigma_\varepsilon^2) + \sigma_\varepsilon^4\sigma_N^2}. \tag{19}$$

Note that $0 < \theta < 1$ is now expressed purely in terms of underlying parameters.

Next, consider $\gamma$. Plug $b$ and $c$ from (17) and (18) into (8), use (9) and manipulate to get:

$$\gamma = \frac{\mu\beta\big[\mu\sigma_U^2 + (1-\mu)\sigma_I^2\big]}{\mu\beta\big[\mu\sigma_U^2 + (1-\mu)\sigma_I^2\big] + \theta\sigma_I^4\sigma_N^2}r. \tag{20}$$

By (20), it is evident that $0 < \gamma < r \leq 1$. This makes the sensitivity parameters $b$ and $c$ clearly positive in (17) and (18), which implies that the rational investors respond parallelly to any shocks in the signal $\widetilde{s}$ and in the noise traders' net demand $\widetilde{N}$.

Finally, substitute (16), (17) and (18) into the tentative model (5), which yields:

$$P = \frac{1}{r}\bigg[\overline{D} - \frac{\sigma_I^2\sigma_U^2}{\mu\sigma_U^2+(1-\mu)\sigma_I^2}A\bigg] + \frac{\mu\beta\sigma_U^2}{r\mu\sigma_U^2+(r-\gamma)(1-\mu)\sigma_I^2}\widetilde{s} + \frac{\sigma_I^2\sigma_U^2}{r\mu\sigma_U^2+(r-\gamma)(1-\mu)\sigma_I^2}\widetilde{N}. \tag{21}$$

Expression (21) describes the formation of the market price of the risky asset as a function of the random variables $\widetilde{s}$ and $\widetilde{N}$. The properties of the model are now clarified to the extent that the effects of marginal changes in the gross yield $r$ on the rational investors' behavior and on the asset market can be studied. Note that, as we analyze the effects of inflation, the change in $r$ is assumedly negative.

## 3. Effects of Inflation

Suppose that there occurs an unforeseen rise in price inflation which depreciates the value of fiat money. To investigate the consequences, consider a marginal reduction in the gross rate $r$ and examine its effects on the rational investors' market decisions and their corollaries to the financial market.

**Proposition 1.** *Inflation does not affect the informed investors' prediction coefficient but makes the uninformed investors' prediction coefficient diminish.*

**Proof.** By formula (3), the informed investors' prediction coefficient reads: $\beta = \frac{\sigma_D^2}{\sigma_D^2+\sigma_\varepsilon^2}$ so that it depends only on underlying parameters, which do not include $r$. Therefore:

$$\frac{\partial\beta}{\partial r} = 0. \tag{22}$$

The effect is zero because in: $E\big[\widetilde{D}\big|\widetilde{s}\big] = \overline{D} + \beta\big[\widetilde{s}\big]$, the prediction coefficient $\beta$ takes into account only the random noise in the private signal $\sigma_s^2 = \sigma_D^2 + \sigma_\varepsilon^2$.

In order to find the respective effect on the uninformed investors' prediction coefficient, take formula (20), differentiate against $r$, recall that $\beta$, $\sigma_I^2$ and $\sigma_U^2 = \theta\sigma_D^2$ are given in terms of underlying parameters by (3), (4) and (19), respectively, and get:

$$\frac{\partial\gamma}{\partial r} = \frac{\mu\beta\left[\mu\sigma_U^2 + (1-\mu)\sigma_I^2\right]}{\mu\beta\left[\mu\sigma_U^2 + (1-\mu)\sigma_I^2\right] + \theta\sigma_I^4\sigma_N^2} > 0. \tag{23}$$

Formula (23) says that a marginal fall in the value of money makes the uninformed investors' prediction coefficient shrink. This is because, by (8), $\gamma = \frac{b\sigma_D^2}{b^2(\sigma_D^2+\sigma_\varepsilon^2)+c^2\sigma_N^2}$ is a parameter that takes into account the sensitivities to the fundament-based signal $\widetilde{s}$ and the sentiment-based actions $\widetilde{N}$, and the variance of the noise traders' net demand $\sigma_N^2$. Note that $0 < \frac{\partial\gamma}{\partial r} < 1$, which says that the prediction coefficient shrinks less compared to the respective proportional fall in $r$. In any case, the result means that the odds to get accurate market information are weakened.

To sum up, (22) and (23) show that inflation does not affect the informed investors' prediction coefficient but makes that of the uninformed investors diminish. □

**Proposition 2.** *Inflation does not affect the risk of rational CARA investors..*

**Proof.** Recalling formula (4), the variance of the informed investors' prediction error reads: $\sigma_I^2 = \frac{\sigma_D^2\sigma_\varepsilon^2}{\sigma_D^2+\sigma_\varepsilon^2}$. Since $\sigma_I^2$ is given in terms of underlying parameters without $r$, the effect is:

$$\frac{\partial\sigma_I^2}{\partial r} = 0. \tag{24}$$

Likewise, by (9) and (19), the variance of the uninformed investors' prediction error reads $\sigma_U^2 = \frac{\mu^2\sigma_\varepsilon^2+\sigma_\varepsilon^4\sigma_N^2}{\mu^2(\sigma_D^2+\sigma_\varepsilon^2)+\sigma_\varepsilon^4\sigma_N^2}\sigma_D^2$, where the underlying parameters do not include $r$, thus:

$$\frac{\partial\sigma_U^2}{\partial r} = 0. \tag{25}$$

The results (24) and (25) reveal that a marginal change in inflation does not affect the variance of the prediction error of neither the informed nor uninformed investors. Moreover, as the risk aversion coefficient equals 1 by the CARA assumption, the above findings mean that any changes in $r$ do not affect rational investors' risk (cf. Caballero and Farhi 2018). □

**Proposition 3.** *Inflation makes market price determination more sensitive to both fundament-based and sentiment-based shocks in market information.*

**Proof.** To examine the effect of a marginal change in $r$ on the sensitivity to the signal $\widetilde{s}$, use formula (17): $b = \frac{\mu\beta\sigma_U^2}{r\mu\sigma_U^2+(1-\mu)(r-\gamma)\sigma_I^2}$, and differentiate against $r$. This produces:

$$\frac{\partial b}{\partial r} = -\frac{\mu\sigma_U^2 + (1-\mu)(1-\frac{\partial\gamma}{\partial r})\sigma_I^2}{\left[r\mu\sigma_U^2 + (1-\mu)(r-\gamma)\sigma_I^2\right]^2}\mu\beta\sigma_U^2 < 0. \tag{26}$$

Expression (26) tells that a reduction in the gross yield $r$ of fiat money strengthens the sensitivity to shocks (that is observed deviations from the expected value zero) in the private signal $\widetilde{s}$.

Likewise, use (18), which says that: $c = \frac{\sigma_I^2 \sigma_U^2}{r\mu\sigma_U^2 + (1-\mu)(r-\gamma)\sigma_I^2}$, and differentiate against $r$ to see the effect on the sensitivity to $\widetilde{N}$:

$$\frac{\partial c}{\partial r} = -\frac{\mu\sigma_U^2 + (1-\mu)(1-\frac{\partial \gamma}{\partial r})\sigma_I^2}{\left[r\mu\sigma_U^2 + (1-\mu)(r-\gamma)\sigma_I^2\right]^2}\sigma_I^2\sigma_U^2 < 0. \tag{27}$$

By (27), a fall in the gross yield $r$ of fiat money strengthens the sensitivity to shocks in the noise traders' net demand $\widetilde{N}$.

Expressions (26) and (27) tell that inflation makes the asset price determination more sensitive to fundament-based shocks in the private signals $\widetilde{s}$ and sentiment-based shocks in noise traders' net demand $\widetilde{N}$. □

**Proposition 4.** *Inflation affects the market price of the risky asset.*

**Proof.** By (5), the market price of the risky asset depends on the random variables $\widetilde{s}$ and $\widetilde{N}$ as: $P = \overline{P} + b\widetilde{s} + c\widetilde{N}$. Take the total differential and write:

$$\frac{dP}{dr} = \frac{\partial \overline{P}}{\partial r} + \frac{\partial b}{\partial r}\widetilde{s} + \frac{\partial c}{\partial r}\widetilde{N}. \tag{28}$$

Concerning the first term on the right-hand side of (28), formula (16) already suggests that a fall in $r$ makes the unconditional expectation $\overline{P}$ rise and vice versa. More precisely,

$$\frac{\partial \overline{P}}{\partial r} = -\frac{1}{r^2}\left[\overline{D} - \frac{\sigma_I^2 \sigma_U^2}{\mu\sigma_U^2 + (1-\mu)\sigma_I^2}A\right] < 0. \tag{29}$$

Recall that very big values of $A$ would make the effect in (29) positive so that a fall in $r$ would make $\overline{P}$ fall. Yet, the quite plausible assumption $\overline{P} > 0$ implies that (29) holds. Moreover, recalling that the expected value of $\widetilde{s}$ and $\widetilde{N}$ is zero and assuming that there are no exogenous shocks in them, the reasonable result is:

$$\frac{dP}{dr} < 0, \tag{30}$$

which says that inflation makes the market price $P$ of the risky asset rise. By (26) and (27), the result holds also when there exist symmetrically positive shocks in $\widetilde{s}$ and $\widetilde{N}$. If the shocks are symmetrically negative or asymmetric, the effect in (28) remains ambiguous in sign. □

**Proposition 5.** *Inflation does not affect the informed investors' demand for the risky asset but affects the uninformed investors' demand.*

**Proof.** By (2), the informed investors' expectation on the random dividend reads: $E\left[\widetilde{D}|\widetilde{s}\right] = \overline{D} + \beta[\widetilde{s}]$, and the effect of a change in the value of money on the informed investors' demand can be derived from (12): $x_I = \frac{E[\widetilde{D}|\widetilde{s}] - rP}{\sigma_I^2}$. The effect reads:

$$\frac{\partial x_I}{\partial r} = -\frac{P + r\frac{\partial P}{\partial r}}{\sigma_I^2}. \tag{31}$$

Concentrate on the nominator on the right-hand side and use formulas (5) and (28) for $P$ and $\frac{\partial P}{\partial r}$, respectively, and write $P + r\frac{\partial P}{\partial r} = (\overline{P} + r\frac{\partial \overline{P}}{\partial r}) + (b + r\frac{\partial b}{\partial r})\widetilde{s} + (c + r\frac{\partial c}{\partial r})\widetilde{N}$. Now, by (16) and (29), the first term on the right-hand side reduces to zero: $(\overline{P} + r\frac{\partial \overline{P}}{\partial r}) = 0$. Moreover, use of (17) and (26) in the second

term yields, after manipulation, $(b + r\frac{\partial b}{\partial r}) = (r\frac{\partial \gamma}{\partial r} - \gamma)\frac{(1-\mu)\mu\beta\sigma_I^2\sigma_U^2}{[r\mu\sigma_U^2 + (1-\mu)(r-\gamma)\sigma_I^2]^2}$, and use of (18) and (27) in the third term produces $(c + r\frac{\partial c}{\partial r}) = (r\frac{\partial \gamma}{\partial r} - \gamma)\frac{(1-\mu)\sigma_I^4\sigma_U^2}{[r\mu\sigma_U^2 + (1-\mu)(r-\gamma)\sigma_I^2]^2}$. Both of these reduce to zero since (20) and (23) straightforwardly show that their joint multiplicator term reduces to zero: $(r\frac{\partial \gamma}{\partial r} - \gamma) = 0$. Therefore, the nominator on the right-hand side of (31) is also zero: $P + r\frac{\partial P}{\partial r} = 0$, which in turn implies that:

$$\frac{\partial x_I}{\partial r} = 0. \tag{32}$$

Expression (32) says that the informed investors' demand for the risky asset is not affected by any changes in the value of money.

The uninformed investors' expectation on the dividend is given by (7) as $E[\widetilde{D}|P] = \overline{D} + \gamma[b\widetilde{s} + c\widetilde{N}]$ saying that the expectation depends on *r* through $\gamma$, *b* and *c*. The uninformed investor' demand is given by (14) as $x_U = \frac{E[\widetilde{D}|P] - rP}{\sigma_U^2}$. The effect of a marginal change in inflation on the uninformed investors' demand then reads:

$$\frac{\partial x_U}{\partial r} = \frac{(b\frac{\partial \gamma}{\partial r} + \gamma\frac{\partial b}{\partial r})\widetilde{s} + \frac{\partial c}{\partial r}\widetilde{N} - (P + r\frac{\partial P}{\partial r})}{\sigma_U^2}. \tag{33}$$

Recall that the last term in the nominator on the right-hand side of (33) is zero: $(P + r\frac{\partial P}{\partial r}) = 0$. The first term in the nominator can be opened by using (17), (20), (23) and (26), and manipulating to get $(b\frac{\partial \gamma}{\partial r} + \gamma\frac{\partial b}{\partial r}) = (r\frac{\partial \gamma}{\partial r} - \gamma)\frac{(1-\mu)\sigma_I^2}{[r\mu\sigma_U^2 + (1-\mu)(r-\gamma)\sigma_I^2]^2\{\mu\beta[\mu\sigma_U^2 + (1-\mu)\sigma_I^2] + \theta\sigma_I^4\sigma_N^2\}}$. Since $(r\frac{\partial \gamma}{\partial r} - \gamma) = 0$, also $(b\frac{\partial \gamma}{\partial r} + \gamma\frac{\partial b}{\partial r}) = 0$. Therefore, only the middle term $\frac{\partial c}{\partial r}\widetilde{N}$ in the nominator of (33) matters so that:

$$\frac{\partial x_U}{\partial r} = \frac{\partial c}{\partial r}\frac{\widetilde{N}}{\sigma_U^2}. \tag{34}$$

Since the expected value of $\widetilde{N}$ is zero, the effect in (4) is zero unless there are exogenous shock deviations from that. Recalling the negative effect in (27), expression (34) says that if there is a positive shock element in the noise traders' random net demand $\widetilde{N}$, a fall in *r* makes the uninformed investors' demand of the risky asset increase. If the shock would be negative, the uninformed investors reduce their demand. In any case, the effect is due to their enforced sensitivity to noise traders' net demand.

Expressions (32) and (34) show that inflation has no effects on the informed investors' demand of the risky asset but makes the uninformed investors' demand change according to noise traders' random transactions in the market. □

**Proposition 6.** *Inflation increases the variance of the risky asset's market price.*

**Proof.** The variance of the market price *P* is calculated like: $var(P) = var(b\widetilde{s}) + var(c\widetilde{N})$, which produces:

$$\sigma_P^2 = b^2(\sigma_D^2 + \sigma_\varepsilon^2) + c^2\sigma_N^2. \tag{35}$$

Taking the total derivative of (35) yields:

$$\frac{d\sigma_P^2}{dr} = 2b\frac{\partial b}{\partial r}(\sigma_D^2 + \sigma_\varepsilon^2) + 2c\frac{\partial c}{\partial r}\sigma_N^2 < 0. \tag{36}$$

Expression (36) says that a fall in the gross yield *r* makes the variance of the market price $\sigma_P^2$ increase. This happens because the uninformed investors get more sensitive to both fundament-based and sentiment-based shocks in their asset demand. In other words, inflation fosters volatility in the financial market. □

## 4. Conclusions

In this paper, we analyzed a Walrasian financial market model with informed and uninformed rational investors and noise traders. In the model, all investors were assumed to trade between riskless and risky holdings in the first period and consume their accumulated wealth in the second period. The aim was to find out the market effects of marginal reductions in the gross yield of the riskless alternative, namely fiat money, which suffers from price inflation. The analysis provided the following, framework-dependent findings:

First, inflation did not affect the informed investors' prediction coefficient but made that of the uninformed investors diminish. This is because the uninformed investors' prediction coefficient includes sentiment-based parameters, which are not included in the informed investors' prediction coefficient. Therefore, only the uninformed investors' probability to get accurate market information was weakened.

Second, it was found that inflation did not affect the variance of the prediction error of either the informed or uninformed rational investors. More importantly, since the CARA risk aversion coefficient was assumed one, the rational investors' risk was found to be independent from inflation. This finding contradicts that of Caballero and Farhi (2018), which is derived in an OLG macro model.

Third, inflation was found to make the rational investors more sensitive to market information consisting of the private signal and the noise traders' random net demand of risky assets. Consequently, increased sensitivity to fundament-based and sentiment-based shocks (that is observed deviations from the expected value zero) affected the market price of the risky asset. Taken that the average market price is positive, the sign of the effect depended on the shocks. In the absence of shocks as well as under symmetrically positive shocks, inflation made the market price rise. Otherwise, the effect remained ambiguous. This finding is in line with Ilomäki and Laurila (2018b).

Fourth, inflation was found to have no effects on the informed investors' demand of the risky asset, but the uninformed investors were found to alter their demand according to the sentiment-based random actions of the noise traders. That is, positive shocks made the uninformed investors' demand increase, and vice versa. This finding is adjacent to that of Ilomäki and Laurila (2018b).

Finally, it was found that inflation increases the variance of the risky asset's market price. This happened because of the uninformed investors' increased sensitivity to both fundament-based and sentiment-based shocks as they base their demand decision on actual market prices. In other words, price volatility in the financial market becomes higher as inflation corrodes the value of money. This theoretical finding is clearer than in Davis and Kutan (2003), which found only weak empirical evidence for it. Overall, the correlation is disputable in the empirical literature (Schwert 1989).

Recalling that the results are interpreted in terms of increasing price inflation, that is as a response to a marginal fall in the gross yield of fiat money, the meaningful effects would be reversed as a response to the more exceptional phenomenon of falling inflation, that is marginal deflation.

Of course, the significance of the theoretical results depends on the acceptability of the model and particularly on the use of the CARA utility function. The function suffers from well-known artificiality, but it is nonetheless widely used in financial studies. Thus, the results of the analysis can be considered as theoretical evidence about the harmfulness of marginal increases in inflation to the financial market, as well as a recommendation for the central banks to commit to inflation targeting.

**Author Contributions:** Conceptualization, H.L. and J.I.; methodology, H.L. and J.I.; software, H.L. and J.I.; validation, H.L. and J.I.; formal analysis, H.L. and J.I.; investigation, H.L. and J.I.; resources, H.L. and J.I.; data curation, H.L. and J.I.; writing—original draft preparation, H.L. and J.I.; writing—review and editing, H.L. and J.I.; visualization, H.L. and J.I.; supervision, H.L. and J.I.; project administration, H.L. and J.I.; funding acquisition, H.L. and J.I. All authors have read and agreed to the published version of the manuscript.

**Funding:** This research received no external funding.

**Conflicts of Interest:** The authors declare no conflict of interest.

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
