# Peer review of "Inflation and Risky Investments"

_jrfm, doi:10.3390/jrfm13120329_

Round 1

Reviewer 1 Report

The paper is well written and interesting, although in my opinion too formulaic and lacking in a more consistent review of the literature.

I believe that the article is susceptible of publication after some small considerations that I am going to do.

My main criticism lies in the fact that the conclusions should indicate whether the results obtained are in line or not with other previously published work.

Furthermore, I would like the authors to further clarify the last paragraph of the conclusions. It is not clear to me what they mean.

Author Response

Dear Referee

We thank you for the insightful comments and respond as follows:

Comment: "The paper is well written and interesting, although in my opinion too formulaic and lacking in a more consistent review of the literature."

Response: We agree that the paper is formulaic and that the literature review is not very consistent, and we have noted that by adding the following paragraph in Introduction:

  • Rows 57-59: As far as we know, our approach has not been previously used to investigate these issues, and we therefore pay close attention to the formalization of the model and to the derivation of the results. For the same reason, there are only few references on which we could directly reflect our findings.     

Comment: "My main criticism lies in the fact that the conclusions should indicate whether the results obtained are in line or not with other previously published work."

Response: Please see the above response. We have also added three references and made the following notes in Conclusions:

  • Row 300: ... framework-dependent ...
  • Row 309: The finding contradicts that of Caballero and Farhi (2018), which is derived in an OLG macro model.
  • Row 316: The finding is in line with Ilomäki and Laurila (2018b). 
  • Row 320: The finding is adjacent with Ilomäki and Laurila (2018b). 
  • Rows 324-325: The finding is clearer than in Davis and Kutan (2003), which finds only weak empirical evidence for it. Overall, the correlation is disputable in the empirical literature (Schwert,1989).   

Comment: "Furthermore, I would like the authors to further clarify the last paragraph of the conclusions. It is not clear to me what they mean."

Response: We have rewritten the last paragraphs in Conclusions as follows:

  • Rows 327-329:Recalling that the results are interpreted in terms of increasing price inflation, that is as a response to a marginal fall in the gross yield of fiat money, the meaningful effects would be reversed as a response to the more exceptional phenomenon of falling inflation, that is marginal deflation.
  • Rows 330-334: Of course, the significance of the theoretical results depends on the acceptability of the model and particularly on the use of the CARA utility function. The function suffers from well-known artificiality, but it is nonetheless widely used in financial studies. Thus, the results of the analysis can be considered as theoretical evidence about the harmfulness of marginal increases in inflation to the financial market, as well as a recommendation for the central banks to commit to inflation targeting. 

Reviewer 2 Report

The paper analyzes the effects of inflation on the rational investors’ behavior and the asset market. The key findings: (1) inflation does not affect the informed investors’ prediction coefficient but makes that of the uninformed investors diminish; (2) inflation does not affect rational investors’ risk but makes the asset price more sensitive to fundament-based and sentiment-based shocks; (3) inflation changes the market price of the risky asset rise, while it has no effects on the informed investors’ demand of the risky asset but affects the uninformed investors’ demand; (4) inflation makes the asset market more volatile. The topic of the manuscript is relevant to the scope of JRFM.

With a few exceptions, the paper is well structured and written.

The introduction states the objectives of the paper.  

The methodology seems sound.

The results and interpretations are correct.  

Major concerns

The authors need to discuss more their results and to link their results to the previous papers.

The authors should highlight the limits of their research. Also, the practical and policy implications of their findings must be stressed.

The paper might be published after some major revisions are made.

Author Response

Dear Referee

We thank you for the insightful comments and reply as follows:

Comment: "The authors need to discuss more their results and to link their results to the previous papers."

Response: We have added three references and made the following notes in Introduction and Conclusions: 

  • Rows 57-59: As far as we know, our approach has not been previously used to investigate these issues, and we therefore pay close attention to the formalization of the model and to the derivation of the results. For the same reason, there are only few references on which we could directly reflect our findings
  • Row 300: ... framework-dependent ...
  • Row 309: The finding contradicts that of Caballero and Farhi (2018), which is derived in an OLG macro model.
  • Row 316: The finding is in line with Ilomäki and Laurila (2018b). 
  • Row 320: The finding is adjacent with Ilomäki and Laurila (2018b). 
  • Rows 324-325: The finding is clearer than in Davis and Kutan (2003), which finds only weak empirical evidence for it. Overall, the correlation is disputable in the empirical literature (Schwert,1989). 

Comment: "The authors should highlight the limits of their research. Also, the practical and policy implications of their findings must be stressed."

Response: We have written the last two paragraphs in Conclusions as follows:  

  • Rows 327-329: Recalling that the results are interpreted in terms of increasing price inflation, that is as a response to a marginal fall in the gross yield of fiat money, the meaningful effects would be reversed as a response to the more exceptional phenomenon of falling inflation, that is marginal deflation.
  • Rows 330-334: Of course, the significance of the theoretical results depends on the acceptability of the model and particularly on the use of the CARA utility function. The function suffers from well-known artificiality, but it is nonetheless widely used in financial studies. Thus, the results of the analysis can be considered as theoretical evidence about the harmfulness of marginal increases in inflation to the financial market, as well as a recommendation for the central banks to commit to inflation targeting.   

Round 2

Reviewer 2 Report

Dear Authors,

I have read the revised version of the paper “Inflation and Risky Investments”. I consider that you have improved significantly the paper according to my recommendations.  Thus, I think that the paper can be published in this form.

Best regards